# A Tool for Air Pollution Scenarios (TAPS v1.0) to enable global, long-term, and flexible study of climate and air quality policies

William Atkinson[1,2], Sebastian D. Eastham[1,3], Y.-H. Henry Chen[1], Jennifer Morris[1], Sergey Paltsev[1], C. Adam Schlosser[1], Noelle E. Selin[2,4]

[1] Joint Program on the Science and Policy of Global Change, Massachusetts Institute of Technology, Cambridge, MA, 02139 USA
[2] Institute for Data, Systems, and Society, Massachusetts Institute of Technology, Cambridge, MA, 02139 USA
[3] Laboratory for Aviation and the Environment, Department of Aeronautics and Astronautics, Massachusetts Institute of Technology, Cambridge, MA, 02139 USA
[4] Department of Earth, Atmospheric, and Planetary Sciences, Massachusetts Institute of Technology, Cambridge, MA, 02139 USA

*Correspondence to*: William Atkinson (watkin@mit.edu) and Noelle E. Selin (selin@mit.edu)

**Abstract.** Air pollution is a major sustainability challenge – and future anthropogenic precursor and greenhouse gas emissions will greatly affect human well-being. While mitigating climate change can reduce air pollution both directly and indirectly, distinct policy levers can affect these two interconnected sustainability issues across a wide range of scenarios. We help to assess such issues by presenting a public Tool for Air Pollution Scenarios (TAPS) that can flexibly construct and assess a variety of climate and air quality emissions pathways through its coupling with socioeconomic modeling of climate change mitigation. In this study, we develop and implement TAPS with three components: recent global and fuel-specific anthropogenic emissions inventories, scenarios of emitting activities to 2100 from the MIT Economic Projection and Policy Analysis model (EPPA), and emissions intensity trends based on recent scenario data from the Greenhouse Gas – Air Pollution Interactions and Synergies (GAINS) model. An initial application shows that in scenarios with less climate and pollution policy ambition, near-term air quality improvements from existing policies are eclipsed by long-term emissions increases – particularly from industrial processes that combine sharp production growth with less stringent pollution controls in developing regions. Additional climate actions would substantially reduce fossil fuel related air pollutant emissions (such as sulfur and nitrogen oxides), while further pollution controls would lead to larger reductions for ammonia and organic carbon. Future TAPS applications could explore diverse regional and global policies that affect these emissions, using pollutant emissions results to drive global atmospheric chemical transport models to study the scenarios' health impacts.

## 1 Introduction

Air pollution is an urgent global health threat, with similar sources to the greenhouse gas (GHG) emissions that drive anthropogenic climate change. Fine particulate matter ($PM_{2.5}$) from fossil fuels and other human sources may have caused millions of premature deaths in recent years (McDuffie et al., 2021; Lelieveld et al., 2019) – while ground-level ozone can exacerbate crop loss and worsen socioeconomic disparities (Saari et al., 2017). Projecting these impacts requires future scenarios for those air pollutants' precursor emissions – but more flexible and accessible tools are needed to elucidate the interdependent but distinct effects of economic, climate, and pollution policy on air quality and human health.

Many research efforts focus on the health "co-benefits" of reduced air pollutant emissions from mitigating GHG emissions (Gallagher and Holloway, 2020; Karlsson et al., 2020). Studies have found that the near-term health benefits from GHG reductions can be on par with or even greater than their near-term climate benefits (Markandya et al., 2018; Shindell et al., 2021). Health benefits vary strongly by region and sector (Vandyck et al., 2020), highlighting the importance of granular analyses and actions that prioritize reductions in high-emitting areas (Polonik et al., 2021). As such, climate action must be complemented by pollution-specific policies to maximize air quality benefits (Reis et al., 2022; Tong et al., 2021) – prompting calls for combined policy assessments to address both issues together (Selin, 2021; Vandyck et al., 2021).

For studies that do vary both climate and air quality policies, most use one of a few existing scenario sets. Current options include the shared socioeconomic pathways (SSPs), a set of global scenarios to 2100 that treat climate and air pollution separately but tie the latter to specific societal narratives (O'Neill et al., 2017; Riahi et al., 2017). Each SSP is associated with a specific pollution control ambition, with regional emissions intensity trends that depend on affluence levels (Rao et al., 2017). These trends were derived from two scenarios developed with the widely used Greenhouse Gas – Air Pollution Interactions and Synergies (GAINS) model: current legislation (CLE), which assumes compliance with existing source- and region-specific emission limits, and the maximum feasible reduction (MFR) case, which assumes gradually increasing application of the lowest-emitting currently available technologies (Amann et al., 2011; Klimont et al., 2017). The resulting air pollutant emission trajectories are included in the sixth Coupled Model Intercomparison Project (CMIP6) and presented online (IIASA, 2018; Rogelj et al., 2018).

Other approaches have a different scope of economic assumptions, timescales, or pollutant species. While several studies vary climate and air quality scenarios across pollutants, they often project emissions intensities based on income rather than policy (Radu et al., 2016; Scovronick et al., 2019). Others have begun to internalize climate-health-economic linkages into optimal policy pathways (Reis et al., 2022), while still using SSP pollution assumptions as baselines. Studies in the Energy Modeling Forum (EMF)-30 use the GAINS scenarios more directly, focusing on black and organic carbon (Smith et al., 2020) or non-agricultural pollutants through 2050 (Vandyck et al., 2018). Since then, GAINS has been updated with more nuanced regions, sectors, and emissions trends (GAINS 4.01 release notes,

2021) – such as recent $SO_2$ (Zheng et al., 2018) and black carbon (Kanaya et al., 2020) reductions in China, as well as revised data and SSP-consistent modeling for the waste management sector (Gomez Sanabria et al., 2021).

Some recent studies have used this updated GAINS model to explore more near-term results or policy extremes. Rafaj et al. (2021) use several integrated assessment models (IAMs) to assess health impacts around current climate policies, proposed policies, or likely attainment of the Paris Agreement's temperature targets (through 2050) – applying GAINS CLE and MFR to the 1.5°C case while maintaining CLE otherwise. Amann et al. (2020) develop a "Clean Air" scenario that includes additional climate, energy, agriculture, and food policies – finding that those additional policies (beyond GAINS' traditional air pollution controls) would lead to nearly double the benefits of reduced $PM_{2.5}$ exposure. Hamilton et al. (2021) use a related scenario of "health in all climate policies", including air pollution reductions, diet change, and active travel benchmarks in nine selected countries. Both these latter papers focus on aggregate effects (comparing base cases to scenarios of those policy levers combined together), and are limited geographically (Hamilton et al., 2021) or temporally to 2040.

We aim to present a more flexible model-based capacity for long-term global scenarios of air pollutant precursor emissions. The resulting Tool for Air Pollution Scenarios (TAPS) can efficiently assess a wide range of climate and air quality policy pathways – from broad to specific at the regional, sectoral, and fuel-based level. In addition, its emissions outputs can readily drive global atmospheric chemical transport models (CTMs) to assess health outcomes – avoiding dependence on previous CTM runs and base years. We demonstrate the tool with illustrative scenarios after coupling with the Economic Projection and Policy Analysis model (EPPA). EPPA is a global multi-region multi-sector recursive–dynamic computable global equilibrium (CGE) model that has been used to study a variety of climate and economic policy impacts (Chen et al., 2015, 2017; Paltsev et al., 2005). While prior efforts have sought to endogenize EPPA's air pollutant emissions trends based on the cost of pollution control options (Sarofim, 2007; Valpergue De Masin, 2003; Waugh, 2012), their use has been limited to select studies (Nam et al., 2013). In contrast, the TAPS framework can be exercised autonomously for flexible scenario development (Fig. 1).

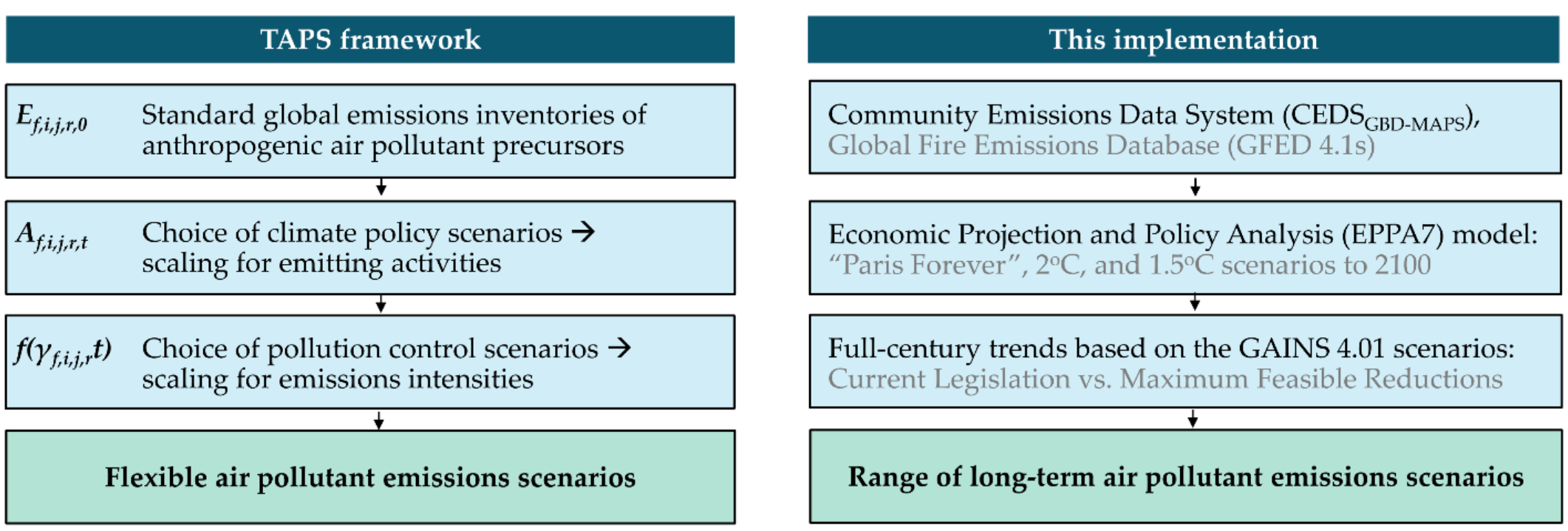

**Figure 1. Summary of the Tool for Air Pollution Scenarios (TAPS) framework and implementation here, based on climate policy scenarios in EPPA7 and pollution control scenarios from the Greenhouse Gas – Air Pollution Interactions and Synergies (GAINS) model. Emissions trends are specific to each fuel *f*, pollutant species *i*, sector *j*, region *r* and time point *t* in the inventories and EPPA7 scenarios used.**

First, we utilize emissions inventories that are well suited for atmospheric modeling work on health impacts – following the SSPs' sources but with updated estimates. Next, we scale those emissions by fuel-specific activities in EPPA, using climate policy scenarios from the global CGE model with full-century time horizons that are longer than most comparable works. Finally, we use updated emissions intensity scenarios from GAINS to assess policies specific to air pollution – while designing pathways that allow for future innovation beyond today's technology options. The following section will describe these steps in turn, before comparing results to SSP benchmarks and discussing next steps for tool refinement and health applications.

## 2 Methodology

Our estimates of air pollutant emissions involve three main inputs: a base-year emissions inventory (Sect. 2.1), a projected trend in energy use and other polluting activities (Sect. 2.2), and a projected trend in emissions intensity (Sect. 2.3). The following equation (based on Fig. 1) summarizes these components:

$$E_{f,i,j,r,t} = E_{f,i,j,r,0} * A_{f,i,r,t} * f(\gamma_{f,i,j,r}t) \quad (1)$$

In this way, the emissions $E_{f,i,j,r,t}$ of inventory fuel $f$, inventory sector $i$, pollutant species $j$, EPPA region $r$, and time $t$ are calculated as the product of base-year emissions $E_{f,i,j,r,0}$, fuel-specific activity $A_{f,i,j,r,t}$, and the function $f(\gamma_{f,i,j,r}t)$ in scenario-specific emissions intensity over time. The below sections discuss each of these components in more detail, as well as the specific scenarios shown in this analysis (Sect. 2.4).

Public versions of the tool, outputs and underlying data are described in the code and data availability section (including processes for figure reproduction). To facilitate coupling with global atmospheric CTMs for health impact analysis, we also include the capability to produce gridded outputs for emissions scaling – following the inventory's spatial distribution as done for the SSPs (Feng et al., 2020). Inputs and Python code can be downloaded and modified to explore the effects of different climate or air quality policies at the region, sector or fuel-based level. While it is simplest to construct scenarios that maintain the structure of current data sources (adjusting from Sect. 2.4), future TAPS applications could theoretically be extended to other inventories or policy model outputs if the database integration steps were completed (adjusting from Sect. 2.1-2.3).

### 2.1 Base-year emissions inventory

This paper uses base-year emissions from the Community Emissions Data System's Global Burden of Disease Major Air Pollution Sources project ($CEDS_{GBD-MAPS}$), an updated version of the anthropogenic air pollutant emissions inventory used in the SSPs as well as atmospheric modeling of health impacts (GEOS-Chem, 2021). CEDS is a global inventory that includes sulfur dioxide ($SO_2$), carbon monoxide (CO), ammonia ($NH_3$), black carbon (BC), organic carbon (OC), nitrogen oxides ($NO_x$), and 23 separate non-methane volatile organic compounds (NMVOC). It offers monthly data globally on a 0.5°×0.5° grid for 1750-2014 (Hoesly et al., 2018), with updates for 1970-2017 (McDuffie et al., 2020) that divide each of 11 sectors into 4 fuel categories (Table A1). Compared to subsequent versions with fewer sectors and no fuel separation, we use the version in McDuffie et al. (2020) because it combines fuel-specific granularity with emissions totals that largely match the latest trends in https://github.com/JGCRI/CEDS (such as lower

BC and OC totals). We use 2014 emissions to match the economic base-year of the GTAP10 database (Aguiar et al., 2019) used in EPPA7 (as described in Sect. 2.2).

We also include emissions of agricultural waste burning, the only type of open burning represented in EPPA's economic activities (Chepeliev, 2020). We follow the SSPs (van Marle et al., 2017) and GEOS-Chem (2021) by using emissions from the Global Fire Emissions Database (GFED) version 4.1s at a 0.25°×0.25° grid (van der Werf et al., 2017). Although GFED gives emissions estimates in terms of dry matter rather than specific pollutants, we use emission factors based on Akagi et al. (2011) to convert these estimates to pollutant-specific emissions, as recommended by GFED and done for the SSPs (see van Marle et al. (2017), Table C1). We use 2014 values to match the base-year inventory of EPPA7, having checked for general consistency with emissions quantities from neighboring years. We do not include emissions from wildfires, non-anthropogenic sources, or other burning sources in GFED (given their lack of representation in EPPA and GAINS). In addition, we do not currently include aviation emissions, given their exclusion from both $CEDS_{GBD\text{-}MAPS}$ and GAINS.

## 2.2 Projecting emitting activities

### 2.2.1 Choice of economic data source

This paper uses full-century activity outputs from several of EPPA's global climate policy scenarios. The latest version of the EPPA model (EPPA7) has 18 regions of the world and 14 economic sectors, as summarized in Appendix B (Paltsev et al., 2021). To scale the base-year emissions inventories by future trends in EPPA, we perform sectoral mapping from each of the 12 inventory sectors (11 from $CEDS_{GBD\text{-}MAPS}$ plus agricultural waste burning from GFED) to one or more of the EPPA7 sectors (Table 1). The process is based on comparisons of CEDS sectors with GTAP10 (Chepeliev, 2020) and its transferal to EPPA sectors, using standard Intergovernmental Panel on Climate Change (IPCC) definitions as a common reference point (Table D1). Since EPPA lacks direct matches for "Waste", "Solvents", or the "Residential" emissions that are often from solid biofuels in CEDS, we use population to scale these sectors. Despite its approximations, this sectoral mapping is useful to keep emissions projections in terms of CEDS and GFED sectors, facilitating SSP comparisons and future atmospheric modeling applications.

### 2.2.2 Choice of activity parameters

Next, we select fuel-specific parameters to scale each emitting activity based on the approach used in the similar U.S. Regional Energy Policy (USREP) model (Yuan et al., 2019). In USREP, emissions from fuel consumption are mostly scaled by future sectoral energy consumption, while non-combustion sources are scaled by that sector's economic output (Dimanchev et al., 2019; Thompson et al., 2014). Here, we apply a similar method to EPPA as described in Table 1, using the four fuel categories (three for combustion, one for "process") in $CEDS_{GBD\text{-}MAPS}$. Each source's scaling is based on the proportion of its base-year emissions (Table A1) as follows:

$$A_{f,i,j,r,t} = \frac{E_{f,i,j,r,0}}{E_{i,j,r,0}} * \sum_{Ei} A_{f,Ei,r,t} \quad , \tag{2}$$

where the EPPA activities $A_{f,Ei,r,t}$ are aggregated via summation across the EPPA sectors *Ei* that are mapped to each inventory sector (see Table 1). For fuel combustion, coal fuels are scaled by EPPA coal energy use trends (in joules), "liquid-fuel-plus-natural-gas" activities are scaled by aggregate oil and gas use trends, and solid biofuel sources are scaled by total sectoral energy use trends. For process-related emissions, some sources like manure management are clearly outside of the energy realm, while others (such as natural gas flaring) may reflect energy activities as well (McDuffie et al., 2020). Accordingly, we scale agricultural waste burning by crop land use trends, and energy or industry "process" sources by their sectors' total energy trends. For agriculture, we use a "per tonne" basis for consistency with GAINS' emissions intensity units – multiplying EPPA's sectoral land use trends (in hectares) by linearly extended production-per-area total crop trends (in tonnes per hectare) from the Food and Agriculture Organization (FAO, 2018). The overall scaling procedure is done for each scenario, pollutant, CEDS or GFED sector, and EPPA region, having linked each CEDS or GFED sector to EPPA sectoral drivers (Table 1) and mapped the CEDS and GFED grids to EPPA regions.

### 2.3 Projecting emissions intensities

Finally, we scale each activity's emissions intensity with region- and sector-specific trends from the GAINS 4.01 scenarios (GAINS 4.01 release notes, 2021; Klimont et al., 2017). Global data and projections from 2000-2050 are available for non-agricultural sectors and air pollutant species through the Energy Modeling Forum (EMF) study scenario data sets (Smith et al., 2020) that have been updated to GAINS 4.01. However, the EMF study does not include $NH_3$, agriculture, or agricultural waste burning. GAINS estimates for these sectors have been provided separately and only for G20 regions. We map both data sets to the CEDS sector-fuel combinations and EPPA regions analyzed here, as described in Table 1, Tables C1-C4**,** and our online repository.

**Table 1. Sectoral mapping and choice of scaling method for each inventory sector.**

| CEDS/GFED sector | EPPA sector(s) | CEDS fuel | EPPA activity | GAINS EMF sector classes |
|---|---|---|---|---|
| **Agriculture** | CROP, FORS, LIVE | Process | Land production | See Table C2-C3 |
| **Agricultural waste** | CROP | Process | Land use | See Table C2-C3 |
| **Energy** | COAL, ELEC, GAS, ROIL | Biofuel | Total energy | Power_Gen_Bio |
| | | Coal | Coal energy | Power_Gen_Coal |
| | | Oil & gas | Oil & gas energy | Power_Gen_(HLF, LLF, NatGas) |
| | | Process | Total energy | Losses, Transformations |
| **Industry** | EINT, FOOD, OTHR | Biofuel | Total energy | End_Use_Industry_Bio |
| | | Coal | Coal energy | End_Use_Industry_Coal |
| | | Oil & gas | Oil & gas energy | End_use_Industry_(HLF, LLF, NatGas) |
| | | Process | Total energy | AACID, CEMENT, CHEMBULK, CHEM, CUSM, NACID, PAPER, STEEL |
| **Commercial** | SERV | Biofuel | Total energy | End_Use_Services_Bio |
| | | Coal | Coal energy | End_Use_Services_Coal |
| **Residential** | Population | Biofuel | Population | End_Use_Residential_Bio |

| | | | | |
|---|---|---|---|---|
| | | Coal | Population | "_Coal |
| | | Oil & gas | Population | "_(HLF, LLF, NatGas) |
| **Other (combustion)** | CROP, FORS, LIVE | Oil & gas | Oil & gas energy | End_Use_Transport_(AGR, OFF)_(LLF, HLF) |
| **Shipping** | TRAN | Oil & gas | Oil & gas energy | "_OFF_(LLF, HLF) |
| **Solvents** | Population | Process | Population | CHEM, CHEMBULK |
| **Transport** | TRAN | Oil & gas | Oil & gas energy | End_Use_Transport_(NatGas, HDT_HLF, HDT_LLF, LDT_HLF, LDT_LLF, MC_LLF) |
| **Non-road transport** | TRAN | Coal | Coal energy | End_Use_Transport_Coal |
| | | Oil & gas | Oil & gas energy | "_(NatGas, OFF_LLF, OFF_HLF) |
| **Waste** | Population | Process | Population | Waste |

**See online repository for full GAINS sector and fuel linkages. CEDS fuel definitions are given in Table S1 of McDuffie et al. (2020) – with bioenergy separated between solid ("Biofuel") and liquid fuels ("Oil & gas"). CEDS-GAINS fuel type discrepancies were recalibrated based on the percent of CEDS fuel emissions covered by GAINS. Residential, Solvents, and Waste sectors were scaled by EPPA population projections, given the lack of sufficient corollary sectors in EPPA. Land production combines land use from EPPA (in area units) with production per area trends from corollary FAO (2018) scenarios. GAINS EMF sectors are given in Table S3 of Rafaj et al. (2021) and https://gains.iiasa.ac.at/models/index.html.**

First, we calculate emissions intensity trends for each GAINS sector by dividing the emissions time series by activity time series. Historical data are available for 2000, 2005, 2010, and 2015 – with projections for the CLE (2020, 2030, 2050) and MFR scenarios (2030, 2050). For missing activity data points, we conduct annual linear interpolation (and/or extension) for sectors with at least two values, or leave emissions intensities constant for sectors with one or no values. For trend extensions that reach zero before 2050, we assume values of zero thereafter. For the GAINS waste sectors – where only emissions (not activities) were given – we assume constant emissions intensities for CLE, versus region-specific trends to zero by 2050 for MFR (based on MFR/CLE emissions ratios) in accordance with a recent GAINS paper (Gomez Sanabria et al., 2021). $NH_3$ waste trends are matched to $NO_x$ due to large data gaps.

For other $NH_3$ sectors, we employ a conservative approach towards estimating intensity reductions outside of the GAINS G20 regions. For MFR, we assume that the non-G20 regions follow the MFR intensity trend of their corollary G20 regions (Table C4) – but with constant intensities in CLE (only following the corollary if its intensity is constant or increasing). For agriculture sectors (where intensity could rise or fall due to shifting land use or dietary patterns), we also incorporate more granular sector trends from the Food and Agriculture Organization's 2050 scenarios of "Business as Usual" (CLE-like) and "Toward Sustainability" (MFR-like), which directly inform the GAINS database as well (FAO, 2018). The resulting intensity trend $I$ combines the GAINS trend ($GI$) with FAO's trend for sector $i$ relative to total production ($F_{r,t}$):

$$I_{f,i,j,r,t} = GI_{f,i,j,r,t} * \frac{F_{i,r,t}}{F_{r,t}} \tag{3}$$

This adjustment allows for the potential of a region's overall agricultural intensity to change based on shifts in the relative share of the emitting sectors within agriculture (such as livestock categories, milk production, or fertilizer tonnage). Associated FAO sectoral and regional mappings are provided in Tables C3-C4.

Next, we prepare the GAINS sectors' emissions intensity trends for integration with EPPA activity trends. First, we scale the trends to a relative value of 1 in EPPA's base-year of 2014, using linear interpolation for the five-year GAINS values. To determine emissions intensity trends by CEDS sector-fuel combination (e.g., Industrial emissions from the "total-coal" fuel), we aggregate the more granular GAINS trends based on the proportion of the sector-fuel's emissions from that GAINS sector – adjusting to the proportion of emissions covered by GAINS in cases where not all the CEDS sector-fuel combinations had a GAINS equivalent. We repeat the process to aggregate from GAINS to EPPA regions.

### 2.4 Implemented scenarios

To illustrate an application of TAPS, we first select three scenarios from EPPA7 to represent variations in climate policy ambition (Table 2), based on Paltsev et al. (2021). The "Paris Forever" scenario assumes the completion of nationally determined contributions (NDCs) from the Paris Agreement (as of March 2021 with more recent adjustments for COVID-19), but no future climate policies beyond those near-term targets. The other two scenarios extend this NDC baseline to the Paris Agreement's long-term temperature goals, using a global emissions cap and price starting in 2030 to provide a 50% chance of limiting warming to 2°C or 1.5°C above pre-industrial levels. (Temperature estimates come from ensemble linkages of the MIT Earth System Model (Sokolov et al., 2018), or MESM, to EPPA's economic results). The 1.5°C scenario features an almost 50% reduction in global greenhouse gas emissions from 2025 to 2030, a highly ambitious projection. As such, these scenarios span a range from current pledges to a much more stringent set of future climate policies.

**Table 2. EPPA7 scenarios analyzed, with selected SSP comparisons.**

| EPPA7 Scenario | Description |
|---|---|
| Paris Forever | Paris Nationally Determined Contribution (NDC) targets (as of March 2021) are met by all countries by 2030 and retained thereafter (Paltsev et al., 2021). |
| Paris 2°C | Same to 2030, with a post-2030 emissions cap, implemented with a global emissions price, to ensure that the 2100 global surface mean temperature does not exceed 2°C above pre-industrial levels with a 50% probability (Paltsev et al., 2021). |
| Paris 1.5°C | Same to 2030, with a post-2030 emissions cap, implemented with a global emissions price, to ensure that the 2100 global surface mean temperature does not exceed 1.5°C above pre-industrial levels with a 50% probability (Morris et al., 2021a). |

| EPPA7 Scenario | RF (W $m^{-2}$) | SSP IAMs compared | RF (W $m^{-2}$) | ΔTemp (°C) | CMIP6 analog |
|---|---|---|---|---|---|
| Paris Forever | 5.95 | RF6.0, Baseline [a] (19) | 5.48-6.43 | 3.23-3.76 | SSP4_60 |
| Paris 2°C | 3.82 | RF3.4 (25) | 3.33-3.57 | 2.13-2.28 | SSP4_34 |
| Paris 1.5°C | 2.87 | RF2.6 (19) | 2.53-2.72 | 1.72-1.82 | SSP1_26 |

**Radiative forcing (RF) and IAM-based temperature change are global mean values for 2100, relative to pre-industrial levels of 1861-1880 in EPPA (Morris, Sokolov, et al., 2021) and 1850-1900 for the SSPs (IIASA, 2018). CMIP6 analog shows the SSP and RF combination that is most similar to each EPPA scenario. [a] IAM scenarios were not included if the radiative forcing (RF) difference from EPPA was greater than 0.5 W $m^{-2}$.**

This range is reflected in the corresponding FAO (2018) scenarios used for agricultural production scaling: "Business As Usual" for "Paris Forever" and "Towards Sustainability" for the 2°C and 1.5°C scenarios. In **Table 2,** we also compare results from each EPPA scenario to CMIP6 scenarios and additional IAM runs from SSPs that have similar

radiative forcing and other assumptions (Feng et al., 2020). While the "SSP5-3.4-Overshoot" scenario does fall in the EPPA forcing ranges, it assumes business-as-usual emissions in the near-term and plentiful negative emissions technologies in the long-term, in contrast to the EPPA scenarios' near-term NDCs and lack of negative emissions.

Turning to pollution control, we use this initial implementation to show the range of outcomes between GAINS CLE and MFR scenarios, based on version 6b of project ECLIPSE (Evaluating the Climate and Air Quality Impacts of Short-Lived Pollutants) as presented by Stohl (2015) and online (IIASA, 2019). After aggregating the GAINS emissions intensity trends to inventory sectors and EPPA regions (Sect. 2.3), we perform exponential fits for all non-constant intensity pathways to enable simpler scenario tuning and harmonization with EPPA's trends out to 2100. This approach also allows for the potential of future innovation beyond today's MFR levels, in contrast to the SSPs' treatment of the current MFR as a "floor" for intensities. (Pathways could differ based on the research question; we describe examples in the discussion and Table 3). Exponentials are designed to pass through base-year values of 1 and MFR waste values of zero for 2050 onward (using uncertainty weightings of 0.01 via Python's scipy curve fitting's sigma parameter). Given the MFR scenario's definition as the maximum feasible pollution reduction, anomalous cases with higher intensities than the corresponding CLE pathway are fixed to CLE levels.

The resulting trends in emissions intensity are reported in the Supplementary Data (before and after exponential fits), with ~5500 trajectories from the 2 GAINS scenarios, 7 pollutants, 18 EPPA regions, and ~20 CEDS sector-fuel combinations. The fit data includes reported $r^2$ values that range from strong (particularly for areas with full data sets such as Western Europe) to weaker in cases with incomplete or abrupt changes in emissions intensities. The trends are highly sector- and region-specific, ranging from sharp decreases (such as 10-100x drops in some transportation cases) to occasional increases (sometimes due to projected fuel switching within the GAINS activities that had been aggregated to the 56 EMF sectors). Increased intensities include CO emissions from steel in Brazil, Africa, and Eastern Europe, as well as $SO_2$ coal emissions from residential (Eastern Europe) and end use industry (Western Europe). Finally, we combine the intensity trends with the linked base-year inventories and revised activity scaling (Eq. 1). Results are presented below and in the online repository, including outputs of all individual emissions trends as well as summary sheets of inventory value, activity scaling, and intensity scaling at notable timepoints (2030, 2050, 2100) for quicker comparisons.

**Table 3: Example emissions intensity trends, based on GAINS scenarios of current legislation (CLE) and maximum feasible reduction (MFR).**

| Scenario | Description |
|---|---|
| **No Improvements** | Assume constant emission factors from base year. |
| **CLE Forever** | Follow CLE emission factors until 2050, and hold them constant afterwards. |
| **CLE Trend Continues** | Fit an exponential function to CLE 2000-2050, and extend that trend to 2100. |
| **Granular Policy Choices** | Adjust CLE trends with regional, sectoral, or fuel-specific policy scenarios. |
| **SSP-like Improvements** | SSP-specific improvements between CLE and MFR, depending on regional income level and reduction stringency of SSP. |
| **MFR Trend Continues** | Fit an exponential function to the historical GAINS data (2000-2015) + MFR scenario (2030-2050), and extend that trend to 2100. |

**For more detailed information on SSP scenarios, see Table 1-2 of the Supporting Information in Rao et al. (2017).**

## 3 Results

### 3.1 Example scenario and SSP comparison

We illustrate an application of TAPS by providing the results for total air pollutant emission trends (Fig. 2), sectoral breakdowns (Fig. 3) and regional breakdowns (Fig. 4). We also compare this implementation to corollary SSP IAM and CMIP6 scenarios (summarized in Table 4). For Fig. 2, we show the full range of SSP-IAM combinations that have a similar radiative forcing to each of the three EPPA-MESM climate scenarios in Table 2. Though the SSPs and EPPA-MESM have slightly different temperature change estimates for a given forcing level, this process represents the closest comparison available between the two data sets. We facilitate this comparison by removing the SSP sectors that are not part of our scaling (aviation and open burning beyond agricultural waste), based on their emissions proportion in the best-fitting CMIP6 scenario (since sectoral non-CMIP6 IAM emissions are not available). This estimate may lead to slight visual differences in SSP data between Fig. 2 (IAM) and Fig. 3 or 4 (CMIP6), but acts as a reasonable first-order comparison with the TAPS scaling.

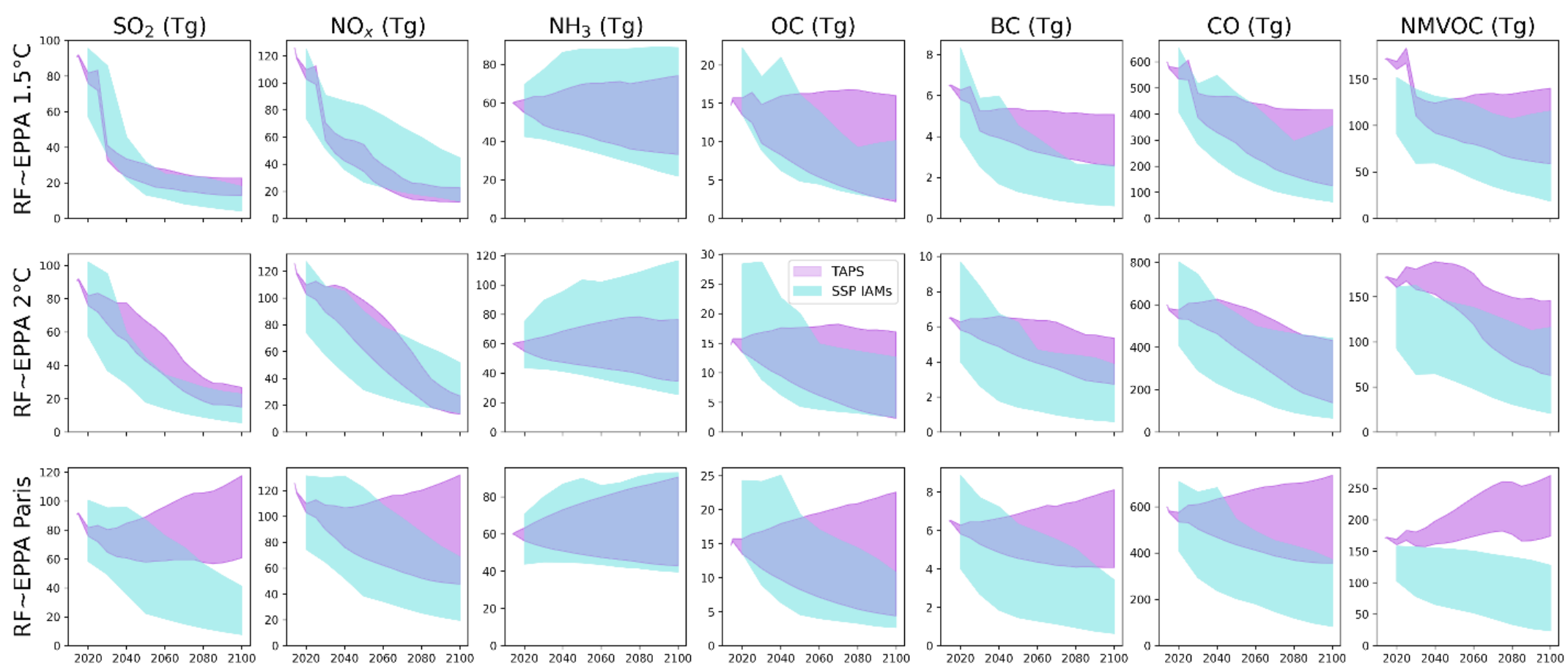

**Figure 2. Global air pollutant emissions trends within the range of GAINS-based scenarios of current legislation (CLE) and maximum feasible reduction (MFR) in Table 3 (purple), as compared to the range of SSP IAM corollaries in Table 2 (blue). IAM estimates are subtracted by sectors not scaled by TAPS (aviation and open burning beyond agricultural waste), based on their emissions proportion in the best-fitting CMIP6 scenario (since sectoral IAM emissions are not available). Quantities of $NO_x$ are in Tg $NO_2$; quantities of BC, OC, and NMVOC are in Tg C.**

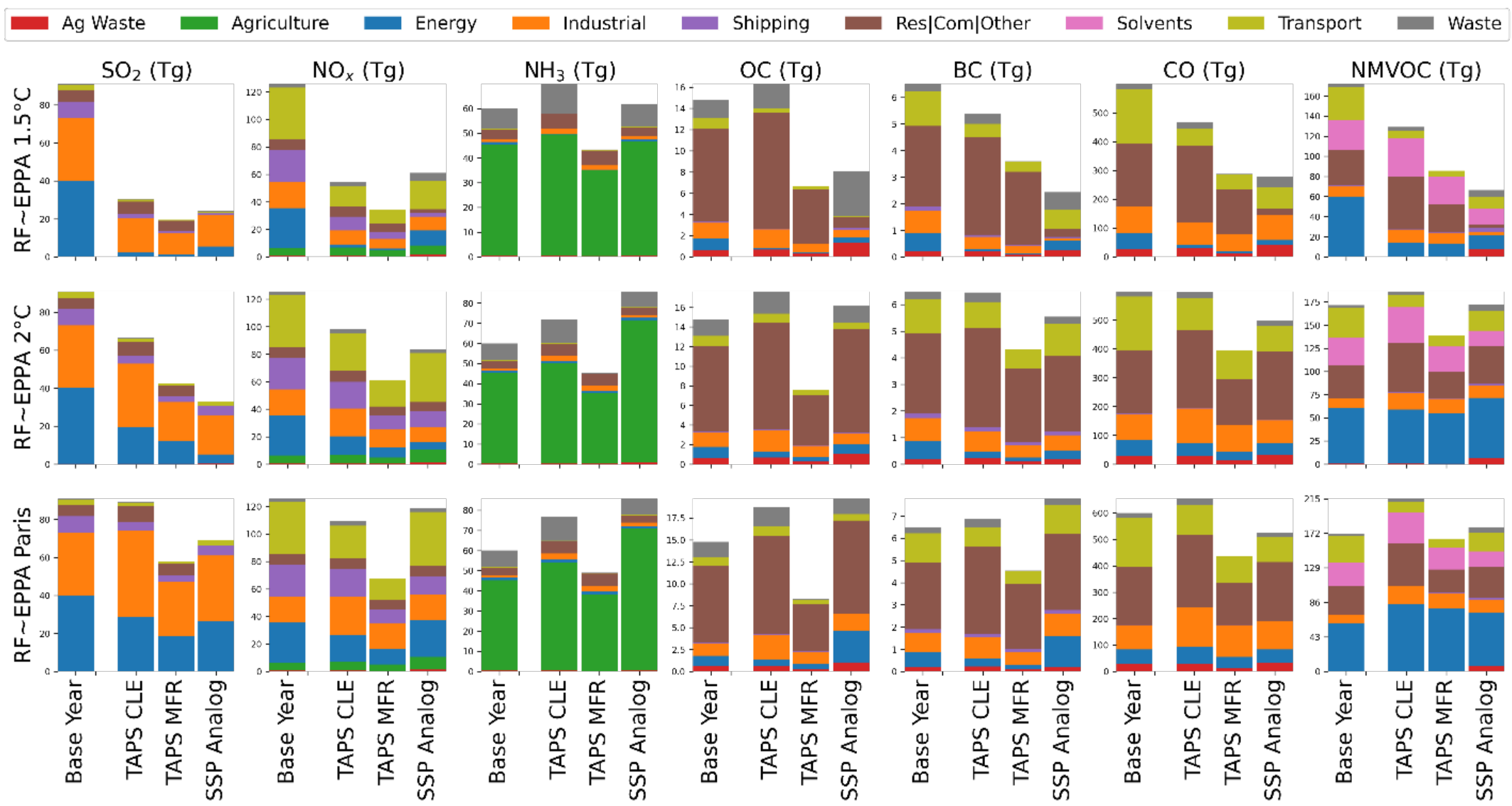

**Figure 3. Sectoral emissions of air pollutants in 2050 under the GAINS-based scenarios of current legislation (CLE) and maximum feasible reduction (MFR) – as compared to the 2014 emissions inventories and corresponding CMIP6 scenarios of SSP1-2.6, SSP4-3.4, and SSP4-6.0 (respectively) for EPPA's 1.5°C, 2°C and Paris Forever scenarios (see Table 2). The 11 $CEDS_{GBD-MAPS}$ sectors (McDuffie et al., 2020) are condensed to the eight in the earlier version used by the SSPs (Hoesly et al., 2018), including the aggregation of residential, commercial, and other combustion ("Res|Com|Other"), plus agricultural waste burning ("Ag Waste") from GFED. Quantities of $NO_x$ are in Tg $NO_2$; BC, OC, and NMVOC are in Tg C.**

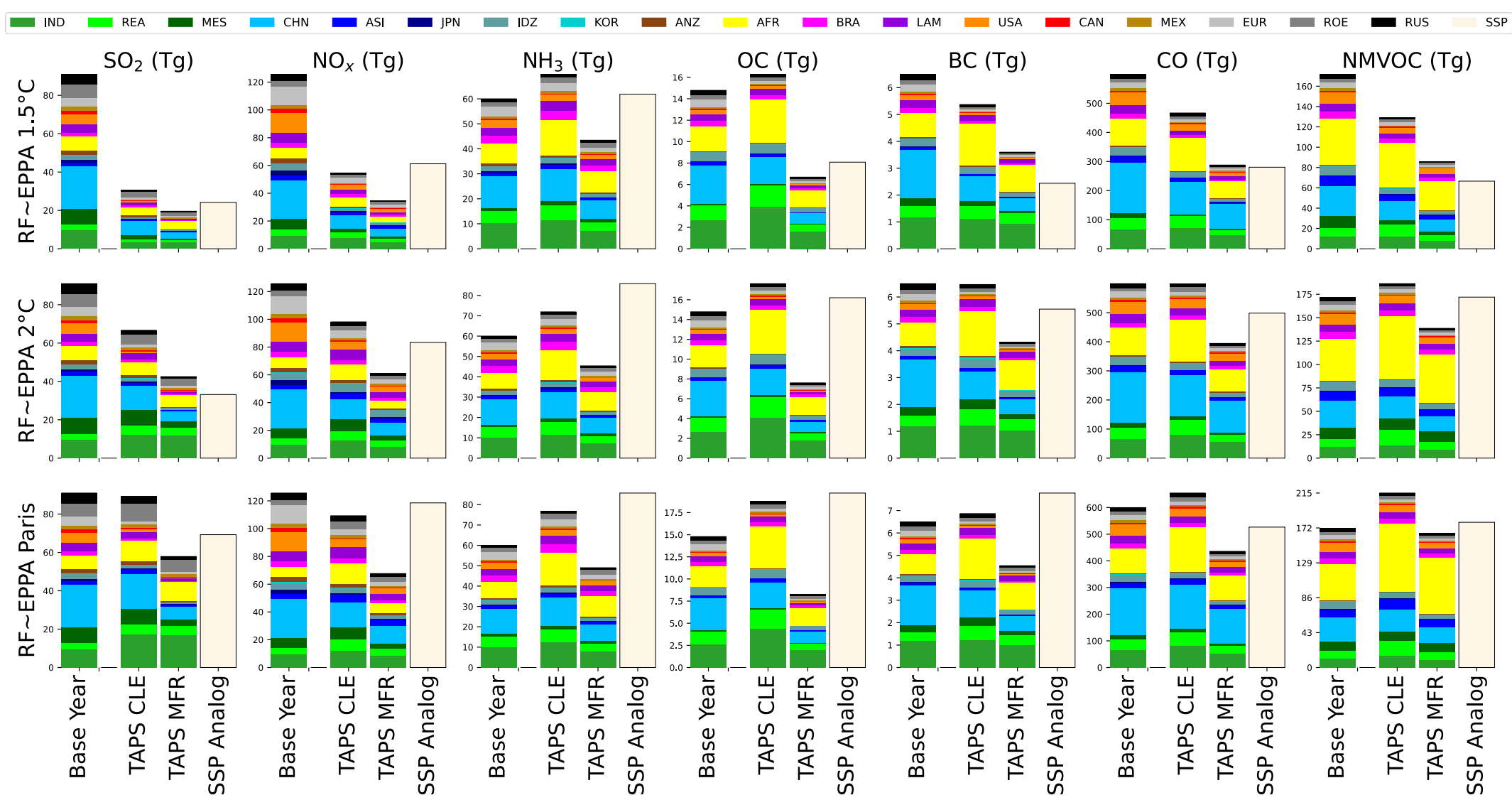

**Figure 4. Regional emissions of air pollutants in 2050 under the GAINS-based scenarios of current legislation (CLE) and maximum feasible reduction (MFR) – as compared to the 2014 emissions inventories and corresponding CMIP6 scenarios of SSP1-2.6, SSP4-3.4, and SSP4-6.0 (respectively) for EPPA's 1.5°C, 2°C and Paris Forever scenarios (as in Table 2). See Table B1 for EPPA region abbreviations. SSP values are shown as global totals due to regional definition discrepancies. Quantities of $NO_x$ are in Tg $NO_2$; BC, OC, and NMVOC are in Tg C.**

**Table 4: Summary of pathways presented.**

| Pathway | Base-Year Emissions | Emitting Activity Scaling | Emissions Intensity Scaling |
|---|---|---|---|
| **TAPS CLE** | 2014; GEOS-Chem 13.0.0 defaults (CEDS, GFED) for anthropogenic emissions | EPPA7 Paris Forever, Paris 2°C, Paris 1.5°C scenarios | Fitted exponential trends from GAINS 4.01 2000-2050 CLE |
| **TAPS MFR** | 2014; GEOS-Chem 13.0.0 defaults (CEDS, GFED) for anthropogenic emissions | EPPA7 Paris Forever, Paris 2°C, Paris 1.5°C scenarios | Fitted exponential trends from GAINS 4.01 2000-2050 MFR |
| **SSP IAMs** | 2005; IAM-specific (Rao et al., 2017) | IAM-specific (Rao et al., 2017) | SSP-based trends via GAINS 3 (Rao et al., 2017) |
| **SSP CMIP6** | 2015; past CEDS (Hoesly et al., 2018) and GFED (van Marle et al., 2017) | IAM-specific (Rao et al., 2017) | SSP-based trends via GAINS 3 (Rao et al., 2017) |

**SSP corollaries from the full range of IAMs are shown in Fig. 2, while sectoral data (Fig. 3) are only available from the CMIP6 subset. For more detailed information on IAM model inputs, see Section 2.2 of the Supporting Information in Rao et al. (2017).**

When comparing initial emissions, IAM inventories differ both in base year (2005 vs. EPPA7's 2014) and emissions values (Fig. 2) – given their variety of sources from the Emissions Database for Global Atmospheric Research (EDGAR) to GAINS to the RCP or even older IPCC inventories (Rao et al., 2017). Even after the inventories have been harmonized in the CMIP6 scenarios (Gidden et al., 2019), their use of an earlier CEDS version (Hoesly et al., 2018) leads to differences such as a base-year OC value that is 30% higher than the updated CEDS value (McDuffie et al., 2020). NMVOC inventories of emissions inside the scope of CEDS are also much lower in the IAMs, especially from the IMAGE and REMIND-MAgPIE models (IIASA, 2018).

In the TAPS example policy scenarios, trajectories do not decrease as often as in the SSPs – showing that emissions could be much higher if emissions intensity improvements are limited to current legislation. While recent studies support these cases of increased emissions (Rafaj et al., 2021), they focus on trends to mid-century. Here, many of the increases are strongest in the late century – implying that any continued improvements in the GAINS-based intensity trends are offset by further increases in activity. This contrast is strongest in industrial "process" emissions sources, where EPPA's sharp increases in activity overpower the slight decreases in emissions intensity. While the full century's trends are shown for context (Fig. 2), the sectoral and regional plots focus on 2050 as the last year with official GAINS scenario data. We next summarize projections for each pollutant category in turn.

### 3.2 Example scenario results by pollutant

In the case of increasing $SO_2$ under EPPA's "Paris Forever" and GAINS' CLE scenarios, continued coal use without desulfurization and/or carbon capture is the primary factor – especially in regions with fewer current pollution controls such as Africa, South Asia, and Eastern Europe. By 2100, the doubling of industrial and residential sector emissions outpaces the decreases in energy and transport sectors. Industrial increases are driven by increased activities (4- to 10-fold by 2100 in those regions) with few intensity improvements, while residential increases are driven by a sharp increase in GAINS-based emissions intensity from Eastern Europe coal use. The GAINS MFR intensities are much lower given the additional pollution controls, halving the industrial emissions compared to CLE and leading to a 3-

fold drop in energy sector emissions by 2100. Still, the increased coal activities of "Paris Forever" (especially in developing areas' non-energy sectors) prevent emissions from decreasing globally, as in Rafaj et al. (2021) but unlike the SSPs. More ambitious climate policy scenarios include rapid declines in coal energy use – leading to declining $SO_2$ emissions even if the intensities of the few remaining emissions sources (mostly industrial and residential) are nonzero.

CO and NMVOC emissions show similar trends. In the case of CO under CLE and "Paris Forever", industrial processes increase in activity (up to 10-fold in India by 2100) as well as intensity for certain regions (4-fold in Africa and 5-fold in Eastern Europe). Pollution controls in MFR reduce these increases, while causing major declines in most other sectors (including residential, unlike with $SO_2$). NMVOC emissions follow these general patterns, with greater influence from energy process sources that have fewer control options in GAINS and more temporal variation from EPPA trends. CLE emissions intensities are relatively flat for energy, industrial, and solvent process sources (with some increases in Brazil and much of Asia), leading to greater emissions under the "Paris Forever" scenario. Further climate policy leads to further declines in energy, transport, and industrial coal, while further pollution policy (in MFR) is more impactful for solvents, residential, and industrial process sources.

Long-term $NO_x$ emissions also increase under less ambitious policies, given the limits of projected intensity improvements in GAINS CLE. In this pathway, increased activities in EPPA lead to increased agriculture and a doubling of industry emissions by 2100 (including a 10-fold increase in India's oil and gas fuel), offsetting initial declines from GAINS intensities and overall reductions in other sectors like energy and transport. The GAINS MFR case gives further intensity reductions, flattening industrial emissions and transitioning energy and transport to near-zero. With further climate policy in the 2°C and 1.5°C scenarios, oil and gas use in EPPA is projected to reach near-zero by late-century as well, leading to lower emissions than most of the IAMs (which may assume less steep energy declines due to their greater reliance on negative emissions).

BC and OC are driven more by residential emissions, which have limited intensity improvements in CLE but much stronger pollution controls in MFR. BC emissions are generally higher than their SSP counterparts, as increased activities overpower intensity improvements for residential, commercial, industrial, and waste sectors. Moving to MFR leads to decreases in all sectors except for commercial, while moving to a 2°C climate scenario reduces energy and industry but not the others. Pollution control actions have an even greater effect for OC. In MFR under "Paris Forever", OC residential and industrial emissions drop 8-fold and 7-fold (respectively) from 2014 to 2100, compared to increases in both sectors under CLE. Across the OC scenarios, adding pollution control ambition leads to more emissions reductions than increasing the climate policy ambition.

$NH_3$ also shows the pronounced effect of pollution control outside of climate policy. In CLE cases, increased agricultural production globally combines with a near-doubled intensity in Africa (by 2100) to offset slight efficiencies elsewhere. When the FAO scenario is changed from "Business as Usual" (CLE-like) to "Toward Sustainability" (MFR-like), the spread of activities is much less emissions-intensive (near-constant in Africa, Eastern Europe, and the Middle East; substantially decreasing elsewhere), and relatively flat land use trends allow for declines in overall

emissions. Non-agricultural $NH_3$ emissions play a smaller role but follow similar patterns, with increased emissions under the limited existing policies and further reductions (such as in waste) under more ambitious policies.

## 4. Discussion

Several factors can help explain the different projection scenarios of TAPS and the SSPs. First, sectoral scaling choices differ between IAMs, as described in Section 2.2 of the Supporting Information in Rao et al. (2017). One example is the much higher value for OC waste emissions in SSP1-2.6 vs. this study (Fig. 3), which comes from a constant-emissions extension of the higher inventory value from the associated IMAGE model (IIASA, 2018). Another difference is the climate policy landscape that has changed between the SSP modeling process (mid-2010s) and the 2021 EPPA scenarios. While the latter may incorporate newer NDC pledges, the SSP IAMs sometimes assumed greater clean energy access and therefore lower biofuel-related BC emissions, for example (IIASA, 2018).

There are also differences between emissions intensity projections in GAINS 3 / ECLIPSE v5a (used by SSPs) and GAINS 4 / ECLIPSE v6b (used here), as the latter includes newer regulatory or technological levers. This is certainly the case for the waste sector, with intensity trends changing from near-constant in GAINS 3 to a net-zero MFR endpoint (elimination of open burning of municipal waste) in GAINS 4 (Gomez Sanabria et al., 2021). More granular regions and sectors, such as the refinement of residential cooking and heating (GAINS 4.01 release notes, 2021), could also affect the pathways where those sectors play major roles (like for black and organic carbon). In addition, the updates reflect the effects of some recent policies, such as the sharp declines of $SO_2$ in China (Zheng et al., 2018).

It is also worth noting the differing structures of each integrated data set in TAPS, particularly with respect to the sectors and regions of CEDS, GFED, EPPA, GAINS, and FAO. The lack of direct EPPA matches for the CEDS sectors of "Residential", "Solvents", and "Waste" necessitates a scaling by population that limits the sectors' range of outcomes. We also make approximations for CEDS' solid biofuel categories, scaling by EPPA's total sectoral energy given the lack of a closer fit. Finally, the regional estimates of $NH_3$ trends beyond the available G20 data (chosen as constant or G20-like intensity paths for each GAINS sector) could be low or high depending on the realities in those areas. Future work could refine these assumptions as improvements become available.

Further application of TAPS could explore other emissions intensity scenarios to inform different research questions (Table 3). This example application demonstrates the range of outcomes between the bounds of a "continued CLE trend" and "continued MFR trend," embodied by the fitted exponentials described above. For other applications, a scenario of constant emission factors could follow other "co-benefits" studies to illuminate air quality benefits from greenhouse gas reductions alone. In addition, a "CLE Forever" case (with emission factors held at the final projected data point) could resemble the "Paris Forever" focus on short-term greenhouse gas policy, while the SSP-like scenarios could be used for more direct comparisons with their income-based pathways. Finally, additional scenario elements such as land use, diet, and active mobility could be incorporated as in recent works – particularly since improving such elements may lead to comparable or even greater health benefits than the pollution-specific policy levers explored here (Amann et al., 2020; Hamilton et al., 2021).

Such scenarios need not be limited to emissions intensity. With the regional, sectoral, and fuel-based EPPA outputs given in the data and code availability, users can readily explore the effects of more granular climate policies applied at those levels. Activity trends could be adjusted to study the effects of sector-specific policies on agricultural land use, fuel-specific policies on coal combustion levels, or region-specific policies that capture individual NDC updates (for example). Given the tool's relatively quick runtime, uncertainty analyses could explore larger ensembles of policy or other inputs to efficiently explore first-order outcome ranges, following the approach of recent EPPA studies on socioeconomic (Morris et al., 2021b) and climate forcing trends (Morris et al., 2021a).

**5. Conclusions**

TAPS provides a flexible and comprehensive model for assessing climate and pollution pathways, integrating recent standard emissions inventories, long-term activity scaling, and scenario-specific emissions intensities. Results from its application to selected scenarios show lower near-term emissions than the SSPs in many cases, both from NDCs' greater climate policy ambition as well as recent pollution reduction actions now captured in GAINS. Less ambitious pathways show increased emissions in the long-term – particularly for the industrial and agricultural processes that have fewer existing controls. These increases are especially pronounced in developing regions where sharply growing activities are combined with fewer planned pollution policies. However, more ambitious climate and pollution policies can curb those increases substantially – from the $SO_2$ and $NO_x$ reductions driven by fuel switching to the $NH_3$ reductions from land use decisions and OC reductions from pollution controls.

Future applications could explore other scenarios by adjusting a range of climate or pollution policy inputs. Assessing other climate or activity scenarios could compare the health impacts of near-term fuel switching versus long-term negative emissions. Additional emissions intensity trends could add the aforementioned elements of land use, diet, or specific innovations beyond today's technological control options. All these scenarios can be applied to specific regions, sectors, or fuels in the framework to explore more granular policies or target short-term actions with high-impact benefits.

Future tool development and linkages could consider other emissions sources – such as aviation, open burning, or wildfires – to explore the futures of additional activities that may be underestimated (Pan et al., 2020) or not fully covered by the default inventories used here. Integration with other modeling tools could examine key inter-pollutant or pollutant-climate feedbacks, such as the increased $NH_3$ emissions rates in a warming world (Yang et al., 2021). External coupling to other ensemble results could address important but out-of-scope elements such as meteorological uncertainty, given its importance in past studies that compared natural variability with other sources of uncertainty in health impacts analysis of air pollution (Pienkosz et al., 2019; Saari et al., 2019).

Finally, additional research with air quality and impact models can assess the health effects of TAPS emissions scenarios as well as their implications for decision-making. Quantified impacts should include a range of mortality and morbidity endpoints to capture recent epidemiological research (Danesh Yazdi et al., 2019), as well as analyses of equity, uncertainty, and sensitivity for key parameters (Hess et al., 2020). Using a combined assessment of climate and pollution policies could help reduce the siloes that have traditionally hindered the consideration of climate-health

linkages in decision-making (Workman et al., 2018). Integrated impact metrics (whether through the weighting of multi-criteria decision analysis or the monetization of benefit-cost analysis) could also inform policy conversations. Ultimately, the TAPS framework could enable more flexible, efficient, and extensive scenario study of policies that affect climate change and health futures.

435 **Appendix A: CEDS reference data**

**Table A1. Percentage of base-year (2014) CEDS emissions from different fuel consumption vs. process sources (broken down by sector, aggregated globally).**

| Sector | Fuel | $SO_2$ | CO | $NH_3$ | BC | OC | NO[a] | $C_2H_4$[b] |
|---|---|---|---|---|---|---|---|---|
| Agriculture | total-coal | 0 | 0 | 0 | 0 | 0 | 0 | 0 |
| | solid-biofuel | 0 | 0 | 0 | 0 | 0 | 0 | 0 |
| | liquid-fuel-plus-natural-gas | 0 | 0 | 0 | 0 | 0 | 0 | 0 |
| | process | 0 | 100 | 100 | 0 | 0 | 0 | 0 |
| Commercial | total-coal | 72 | 0 | 25 | 44 | 49 | 52 | 24 |
| | solid-biofuel | 1 | 0 | 27 | 49 | 25 | 11 | 27 |
| | liquid-fuel-plus-natural-gas | 27 | 100 | 48 | 7 | 26 | 38 | 50 |
| | process | 0 | 0 | 0 | 0 | 0 | 0 | 0 |
| Energy | total-coal | 64 | 51 | 5 | 7 | 3 | 10 | 0 |
| | solid-biofuel | 0 | 3 | 2 | 37 | 9 | 1 | 0 |
| | liquid-fuel-plus-natural-gas | 19 | 32 | 7 | 1 | 2 | 8 | 0 |
| | process | 17 | 14 | 87 | 55 | 86 | 81 | 100 |
| Industry | total-coal | 45 | 55 | 5 | 21 | 54 | 43 | 28 |
| | solid-biofuel | 0 | 9 | 38 | 74 | 20 | 8 | 26 |
| | liquid-fuel-plus-natural-gas | 20 | 32 | 10 | 6 | 26 | 5 | 8 |
| | process | 35 | 5 | 47 | 0 | 0 | 44 | 38 |
| Non-road transport | total-coal | 0 | 0 | 0 | 0 | 0 | 0 | 0 |
| | solid-biofuel | 0 | 0 | 0 | 0 | 0 | 0 | 0 |
| | liquid-fuel-plus-natural-gas | 100 | 100 | 100 | 100 | 100 | 100 | 100 |
| | process | 0 | 0 | 0 | 0 | 0 | 0 | 0 |
| Other | total-coal | 38 | 1 | 12 | 23 | 13 | 10 | 6 |
| | solid-biofuel | 0 | 2 | 9 | 43 | 8 | 20 | 16 |
| | liquid-fuel-plus-natural-gas | 62 | 97 | 79 | 34 | 79 | 70 | 78 |
| | process | 0 | 0 | 0 | 0 | 0 | 0 | 0 |
| Residential | total-coal | 70 | 8 | 0 | 8 | 13 | 13 | 3 |
| | solid-biofuel | 20 | 58 | 97 | 92 | 70 | 87 | 96 |
| | liquid-fuel-plus-natural-gas | 10 | 33 | 3 | 0 | 17 | 1 | 1 |
| | process | 0 | 0 | 0 | 0 | 0 | 0 | 0 |
| Shipping | total-coal | 0 | 0 | 0 | 0 | 0 | 0 | 0 |
| | solid-biofuel | 0 | 0 | 0 | 0 | 0 | 0 | 0 |

| | | | | | | | | |
|---|---|---|---|---|---|---|---|---|
| | liquid-fuel-plus-natural-gas | 100 | 100 | 100 | 100 | 100 | 100 | 100 |
| | process | 0 | 0 | 0 | 0 | 0 | 0 | 0 |
| Solvents | total-coal | 0 | 0 | 0 | 0 | 0 | 0 | 0 |
| | solid-biofuel | 0 | 0 | 0 | 0 | 0 | 0 | 0 |
| | liquid-fuel-plus-natural-gas | 0 | 0 | 0 | 0 | 0 | 0 | 0 |
| | process | 0 | 0 | 100 | 0 | 0 | 0 | 0 |
| Transport | total-coal | 0 | 0 | 0 | 0 | 0 | 0 | 0 |
| | solid-biofuel | 0 | 0 | 0 | 0 | 0 | 0 | 0 |
| | liquid-fuel-plus-natural-gas | 100 | 100 | 100 | 100 | 100 | 100 | 100 |
| | process | 0 | 0 | 0 | 0 | 0 | 0 | 0 |
| Waste | total-coal | 0 | 0 | 0 | 0 | 0 | 0 | 0 |
| | solid-biofuel | 0 | 0 | 0 | 0 | 0 | 0 | 0 |
| | liquid-fuel-plus-natural-gas | 0 | 0 | 0 | 0 | 0 | 0 | 0 |
| | process | 100 | 100 | 100 | 100 | 100 | 100 | 100 |

**[a] CEDS reports $NO_x$ as NO and NMVOC as speciated compounds; [b] $C_2H_4$ is shown as an example NMVOC species. Other NMVOC species may show differences, such as more "process" emissions from solvents. Global aggregate proportions are shown here for context; full regional and speciated values are available at our online repository. CEDS fuel definitions are given in Table S1 of McDuffie et al. (2020), with bioenergy separated between solid and liquid fuels.**

## Appendix B: EPPA7 reference definitions

**Table B1. EPPA7 regions and sectors, as described in Paltsev (2021).**

| Region code | Region name | Sector code | Sector name |
|---|---|---|---|
| AFR | Africa | COAL | Coal |
| ANZ | Australia, New Zealand & Oceania | CROP | Agriculture - Crops |
| ASI | East Asia | DWE | Ownership of Dwellings |
| BRA | Brazil | EINT | Energy-Intensive Industries |
| CAN | Canada | ELEC | Electricity |
| CHN | China | FOOD | Food |
| EUR | European Union+ | FORS | Agriculture - Forestry |
| IDZ | Indonesia | GAS | Gas |
| IND | India | LIVE | Agriculture - Livestock |
| JPN | Japan | OIL | Crude Oil |
| KOR | South Korea | OTHR | Other |
| LAM | Latin America | ROIL | Refined Oil |
| MES | Middle East | SERV | Services |
| MEX | Mexico | TRAN | Transport |
| REA | Rest of Asia | | |
| ROE | Eastern Europe and Central Asia | | |
| RUS | Russia | | |
| USA | USA | | |

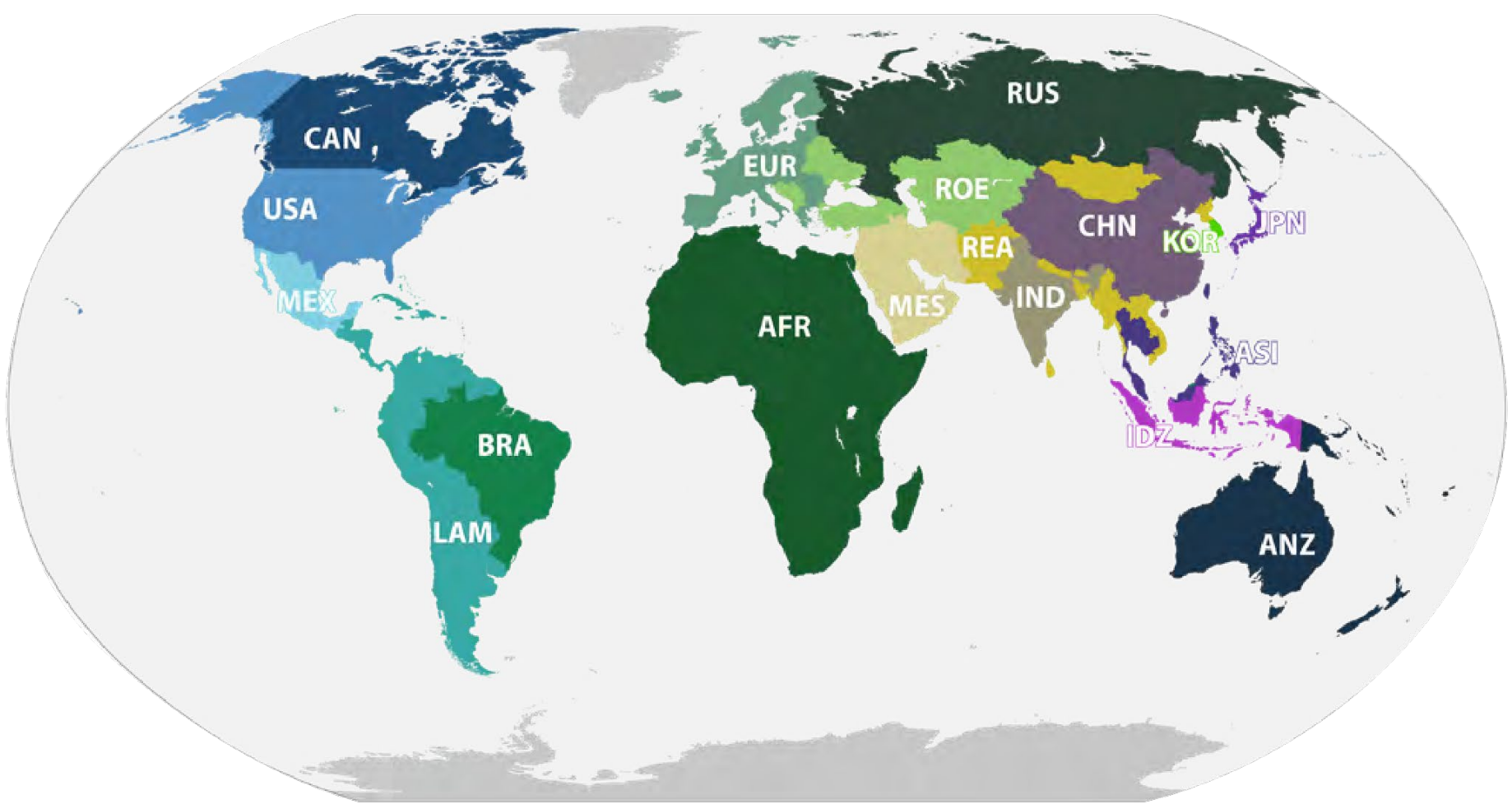

**Figure B1. Map of EPPA7 regions of the world, from Paltsev (2021) with reproduction rights granted.**

## Appendix C: Mapping from GAINS model

**Table C1. Mapping from GAINS EMF (based on IMAGE) to EPPA7 regions.**

| EPPA7 | GAINS EMF | EPPA7 | GAINS EMF | EPPA7 | GAINS EMF |
|---|---|---|---|---|---|
| **CAN** | 1 Canada | **AFR** | 10 South Africa | **IND** | 18 India |
| **USA** | 2 USA | **EUR** | 11 Western Europe | **KOR** | 19 Korea |
| **MEX** | 3 Mexico | **EUR** | 12 Central Europe | **CHN** | 20 China+ |
| **LAM** | 4 Rest Central America | **ROE** | 13 Turkey | **ASI** | 21 Southeastern Asia |
| **BRA** | 5 Brazil | **ROE** | 14 Ukraine+ | **IDZ** | 22 Indonesia+ |
| **LAM** | 6 Rest South America | **ROE** | 15 Asia-Stan | **JPN** | 23 Japan |
| **AFR** | 7 Northern Africa | **RUS** | 16 Russia+ | **ANZ** | 24 Oceania |
| **AFR** | 8 Western Africa | **MES** | 17 Middle East | **REA** | 25 Rest South Asia |

**IMAGE regions are given in Fig. S7.1 of Klimont et al. (2017) and compared to Fig. 2. Regions in blue differ slightly from EPPA definitions.**

**Table C2. Mapping from GAINS $NH_3$ to CEDS/GFED inventory sectors and fuels.**

| Inventory sector | CEDS fuel | GAINS $NH_3$ sector classes | GAINS $NH_3$ sector class names |
|---|---|---|---|
| **Ag. waste burning** | Process | WASTE_AGR | Agricultural waste burning |
| **Agriculture** | Process | AGR, COWS, FCON, FERTPRO | Livestock and fertilizer (Table C3) |
| **Energy** | Coal | PP - BC1, BC2, DC, HC1, HC2, HC3 | Power plants (brown, derived, and hard coal) |
| | Biofuel | PP - OS1, OS2 | " (biomass and waste fuels) |
| | Oil & gas | PP - GAS, GSL, HF, LPG, MD | " (natural gas, gasoline, heavy fuel oil, liquified petrol gas, diesel) |
| | Process | CON, PROD_AGAS, WASTE_FLR | Conversion, flaring and venting |
| **Industry** | Coal | IN_OC - BC1, BC2, DC, HC1, HC2, HC3 | Industrial (brown, derived, hard coal) |
| | Biofuel | IN_OC - OS1, OS2 | " (biomass and waste fuels) |
| | Oil & gas | IN_OC - GAS, GSL, HF, LPG, MD | " (natural gas, gasoline, heavy fuel oil, liquified petrol gas, diesel) |
| | Process | IN_BO, IO_NH3_EMISS | Boiler and other emissions |
| **Residential, Commercial** | Coal | (DOM) - BC1, BC2, DC, HC1, HC2, HC3 | Residential-commercial (brown/derived/hard coal) |
| | Biofuel | (DOM) - OS1 | " (biomass) |
| | Oil & gas | (DOM) - GAS, GSL, HF, LPG, MD | " (natural gas, gasoline, heavy fuel oil, liquified petrol gas, diesel) |
| **Other (combustion)** | Oil & gas | TRA_OT_(AGR, CNS, LB, LD2) | Off-road engines, mopeds, construction & agriculture vehicles |
| **Shipping** | Oil & gas | TRA_OTS | Maritime |
| **Solvents** | Process | IO_NH3_EMISS | Other industrial $NH_3$ emissions |
| **Transport** | Oil & gas | TRA_RD | All road transportation |
| **Non-road transport** | Oil & gas | TRA_OT_INW, TRA_OT_RAI | Inland waterways, railways |
| **Waste** | Process | WT_NH3_EMISS[a] | Trash burning |

**See full table (with a row for each of the 198 GAINS $NH_3$ sectors) in Supplementary Data. CEDS fuel definitions are given in Table S1 of McDuffie et al. (2020) – with bioenergy separated between solid ("Biofuel") and liquid fuels ("Oil & gas"). Comparisons are based on Table S3 in Rafaj et al. (2021), with sectoral abbreviations described further in GAINS Online. [a]Since $NH_3$ "Waste" data were only available for two countries, emissions intensity trends follow $NO_x$ "Waste" trends based on Gomez Sanabria et al. (2021).**

**Table C3. Mapping from GAINS agricultural sectors to FAO activities.**

| GAINS | FAO |
|---|---|
| **AGR_BEEF** | Beef and veal |
| **AGR_COWS** | Raising of cattle |
| **AGR_OTANI-BS** | Raising of buffaloes |
| **AGR_OTANI-CM, -FU, -HO** | Raising of livestock (total) |
| **AGR_OTANI-SH** | Raising of sheep |
| **AGR_PIG** | Raising of pigs |

| | |
|---|---|
| **AGR_POULT** | Raising of poultry |
| **COWS_3000_MILK** | Raw milk |
| **FCON, FERTPRO** | NPK_consumption |

**Based on GAINS sector abbreviations at https://gains.iiasa.ac.at/models/index.html and FAO sectors in regional aggregate data.**

**Table C4. Mapping from $NH_3$ data sources to EPPA7 regions.**

| EPPA7 | G20 Corollary | FAO Corollary |
|---|---|---|
| **CAN** | USA | High-income |
| **USA** | USA | High-income |
| **MEX** | Mexico | Latin America/Caribbean |
| **LAM**[b] | Argentina | Latin America/Caribbean |
| **BRA** | Brazil | Latin America/Caribbean |
| **AFR**[b] | South Africa | Sub-Saharan Africa |
| **EUR** | United Kingdom; France; Germany | High-income |
| **ROE**[b] | Turkey | Europe/Central Asia |
| **RUS** | Russia[a] | Europe/Central Asia |
| **MES**[b] | Turkey | Near East/North Africa |
| **IND** | India[a] | South Asia |
| **KOR** | South Korea[a] | EAP excluding China |
| **CHN** | China[a] | China |
| **ASI**[b] | China[a] | EAP excluding China |
| **IDZ**[b] | China[a] | EAP excluding China |
| **JPN** | Japan[a] | EAP excluding China |
| **ANZ** | Australia | High-income |
| **REA**[b] | India[a] | South Asia |

**Full GAINS data were only provided for G20 regions. Countries that approximate other regions are shown in blue, while corollaries that represent a part of their EPPA regions (or vice versa) are in purple. FAO regions are shown in Fig. 1.2 of FAO (2018). [a] Countries with subnational regions in GAINS were aggregated based on their proportional emissions. [b] Scaling for EPPA regions not well-captured by the GAINS G20 coverage is described in Sect. 2.3.**

## Appendix D: IPCC sectoral references

**Table D1. IPCC sectoral definitions for EPPA scaling of sectors from the chosen emissions inventories.**

| IPCC code | Activity | CEDS sector | EPPA sectoral scaling |
|---|---|---|---|
| **3** | Agriculture process emissions | Agriculture | CROP, FORS, LIVE |
| **4F** | Agricultural waste burning | N/A; from GFED | CROP |
| **1A1** | Electricity/fuel production | Energy | COAL, ELEC, GAS, ROIL |
| **1B** | Fugitive fuel emissions | Energy | COAL, ELEC, GAS, ROIL |
| **7A** | Fossil fuel fires | Energy | COAL, ELEC, GAS, ROIL |
| **1A2** | Industrial combustion | Industry | EINT, FOOD, OTHR |
| **1A5** | Other industrial (combustion) | Industry | EINT, FOOD, OTHR |
| **2A-2C, H, L** | Industrial process emissions | Industry | EINT, FOOD, OTHR |
| **6A** | Other industrial (process) | Industry | EINT, FOOD, OTHR |
| **1A4a** | Commercial/institutional | Commercial | SERV |
| **1A4b** | Residential | Residential | Population |
| **1A4c** | Other combustion | Other (combustion) | CROP, FORS, LIVE |
| **1A3d(i)** | International shipping, oil tankers | Shipping | TRAN |
| **2D** | Solvents | Solvents | Population |
| **1A3,1C** | Aviation | N/A | |
| **1A3b** | Road transportation | Transport | TRAN |
| **1A3c** | Rail transportation | Non-road transport | TRAN |
| **1A3d(ii)-e(ii)** | Domestic navigation, other transport | Non-road transport | TRAN |
| **5** | Waste/wastewater emissions | Waste | Population |

**Inventory versions include $\text{CEDS}_{\text{GBD-MAPS}}$ (McDuffie et al., 2020) for most anthropogenic emissions, as well as GFED4.1s (van der Werf et al., 2017) for biomass burning. Since only agricultural waste burning is included in EPPA through GTAP/EDGAR, other sources of burning emissions are not scaled by EPPA outputs. Aviation was not scaled in this work due to its exclusion from both $\text{CEDS}_{\text{GBD-MAPS}}$ and GAINS. "Other combustion" includes sources from agriculture, forestry, and fishing. Sectoral scaling from EPPA largely reflects the distribution of activities in GTAP10 / EDGAR5.0 sectors (Chepeliev, 2020), which are then mapped to representative EPPA7 sectors.**

## Code and data availability

A (frozen) version of the tool code, processing scripts, data outputs, figure production, and any inputs not described below can be found on Zenodo at https://doi.org/10.5281/zenodo.6452104 (Atkinson et al., 2022). The current version can be found on Github at https://github.com/watkin-mit/TAPS, including the full user manual (https://github.com/watkin-mit/TAPS/wiki) and open-source MIT license. Input data are available as follows:

- $CEDS_{GBD-MAPS}$ (anthropogenic emissions inventory): accessed through GEOS-Chem at http://ftp.as.harvard.edu/gcgrid/data/ExtData/HEMCO/CEDS/v2020-08/, DOI:10.5281/zenodo.3754964 (McDuffie et al., 2020)
- GFED4.1s (agricultural waste burning inventory): accessed through GEOS-Chem at http://ftp.as.harvard.edu/gcgrid/data/ExtData/HEMCO/GFED4/v2015-10/, DOI:10.22033/ESGF/input4MIPs.10455, with dry matter emission factors from http://www.globalfiredata.org/ar6historic.html (van Marle et al., 2017)
- EPPA7 scenario data (last accessed 7 May 2021): see the above DOI, with further information at https://globalchange.mit.edu/research/research-tools/human-system-model (Paltsev et al., 2021)
- GAINS 4.01 scenario data (last accessed 12 October 2021): https://gains.iiasa.ac.at/models/ (IAM resolution, ECLIPSE v6b CLE and MFR, EMF30 resolution with G20 GAINS sectors for $NH_3$) available with a free account (Amann et al., 2011; GAINS 4.01 release notes, 2021; Klimont et al., 2017; Smith et al., 2020)
- FAO scenario data (last accessed 21 January 2022): https://www.fao.org/global-perspectives-studies/food-agriculture-projections-to-2050/en/ (FAO, 2018)
- SSP IAM comparisons (last accessed 30 April 2021): Version 2.0, DOI:110.1016/j.gloenvcha.2016.05.009 (Riahi et al., 2017) via the SSP database: https://tntcat.iiasa.ac.at/SspDb/
- SSP CMIP6 comparisons (last accessed 30 April 2021): Version 2.0, DOI:10.5194/gmd-12-1443-2019 (Gidden et al., 2019) via the SSP database: https://tntcat.iiasa.ac.at/SspDb/
- Global population distribution (last accessed 28 October 2020): Gridded Population of the World, Version 4.11, Population Count Adjusted to Match 2015 Revision of UN WPP Country Totals, https://doi.org/10.7927/H4PN93PB (CIESIN, 2018)

## Author contribution

WA compiled the data sources, led the code development, and prepared the paper with contributions from NES, CAS, SE, JM, SP, and YHC. Project oversight came from NES, CAS, and SE, with data and feedback from YHC, JM, and SP.

## Competing interests

The authors declare that they have no conflict of interest.

## Acknowledgements

This work was carried out with support from the U.S. EPA and its Science to Achieve Results (STAR) program (no. R834279). The research has not been subject to EPA review and therefore does not necessarily reflect its views; no official endorsement should be inferred. Use of the EPPA model was made possible by the MIT Joint Program on the Science and Policy of Global Change, which is supported by an international consortium of government, industry and foundation sponsors (a list can be found at: https://globalchange.mit.edu/sponsors/current). Research was also supported by the Biogen Foundation, MIT's Leading Technology and Policy Initiative, and its Research to Policy Engagement Initiative. We also thank Zbigniew Klimont for providing comments on an earlier draft, as well as for modeling inputs with the help of Robert Sander and Shilpa Rao.

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
