# Peer review of "A Tool for Air Pollution Scenarios (TAPS v1.0) to enable global, long-term, and flexible study of climate and air quality policies"

_Geoscientific Model Development, 2022_

## Referee Comment (RC2)

Review comments

In this manuscript, the authors build a framework to study the long-term emissions of GHGs and major air pollutants by coupling with other socioeconomical models. This tool provides flexible to study the wide range of emission changes under various climate scenarios. The methodology is reasonable, and potential be useful. However, I have several concerns on how the authors solve their discrepancies between their framework with SSPs, and how the other researchers could be beneficial from this tool.

**Methods:**

To make this tool usable or accessible for other users, I want the authors to add their suggestions on how to include the fire & aviation emissions which are currently not included in this framework (line 142-144).

As acknowledged in their discussions, the authors also discussed the discrepancies between TAPS and SSPs. To make community adopt their framework for further climate and air quality related research, I wonder how the authors can convince the users that these discrepancies between TAPS and SSPs are acceptable.

**Minor comments:**

Line 32: delete "anthropogenic".

Line 40-41: I suggest the authors to expand their references on co-benefits analysis. These two included in the paper are not very representative.

Line 83-85: "dependence on previous CTM runs and base years"

Line 90: the authors discussed that the usages of EPPA are quite "limited" in previous studies. I wonder then why the authors choose the EPPA not other IAMs? Any potential limitations for potential users to adopt this product?

Line 94: describe "EPPA7"

Line 125: there is no link or doi for such reference "GEOS-Chem, 2021".

Line 142: I am very curious to see the authors' check on the consistency between EPPA7 and GFED.

---

## Author Comment (AC1)

Author response to reviews for "A Tool for Air Pollution Scenarios (TAPS v1.0) to enable global, long-term, and flexible study of climate and air quality policies"
By William Atkinson et al.

We first thank the reviewers for their time and valuable comments, which have improved the quality, nuance, and completeness of the manuscript.

We now respond to the reviewer comments, which are reproduced in black text below. Our responses follow immediately in red text, and any additions to the manuscript are included in italic red text, along with line references, which refer to their locations in the revised manuscript.

Anonymous Referee #1

**General comments**

The study describes a modelling framework to estimate future air pollutant emission pathways for alternative scenarios that combine different climate change mitigation strategies with air pollution controls. The incorporation of air quality and the potential co-effects to global scenario analysis is at the heart of scientific and policy analysis, and the presented TAPS framework will be a relevant contribution to the area. Particularly, the decomposition of future emissions into activity variations and emission intensity changes is helpful to understand future emission trends and support policymakers in policy design.

I have some general comments. First, I believe that the contribution of the study could be re-articulated. The generation of alternative scenarios of air pollutant precursor emissions has already been addressed by different models in the context of global scenario analysis. I think the main innovation of the presented TAPS framework is the flexibility to allow the user to combine alternative climate objectives with air pollution control policies, using the most updated emission factors from GAINS. This should be more clearly stated in order to better frame the study. Also, I include some specific comments (below) aimed at clarifying the contribution of the work.

My second general comment is associated with the design of the air pollution control scenarios (Table 3). While the authors fit an exponential function to extend the EF assumptions from GAINS (2050) to the end of the century, it is not clear enough what is the rationale behind extending the MFR approach. According to the definition, MFR represent the "maximum technically feasible reduction" based on today's knowledge of technological capacity. Therefore, having an additional decreasing emission intensity for the second part of the century is very uncertain and should be justified. Considering this uncertainty, adding an analogous "MFR forever" scenario could be useful.

Overall, the study is a relevant contribution for a large research community focused on global scenario analysis, and I would recommend it for publication after moderate revisions.

We appreciate the thorough review, with many suggestions to improve the paper's completeness. Below are our responses to the two general comments:

On the contribution of the study, we welcome the reviewer's rephrasing of our contribution, as it matches well with our goals. We have restated key sentences in the abstract and introduction to more clearly communicate the contribution of the study:

*L17-19: We help to assess such issues by presenting a public Tool for Air Pollution Scenarios (TAPS) that can flexibly assess pollutant emissions from a variety of climate and air quality actions, through the tool's coupling with socioeconomic modeling of climate change mitigation.*

*L84-86: We aim to present a more flexible model-based capacity for long-term global scenarios – allowing the user to specify diverse levels of climate actions and pollution controls to estimate their combined effect on air pollutant precursor emissions.*

In other sections of the manuscript, we have reviewed language for consistency with this message.

On the second general comment, we agree that the suggested scenario would be useful (though post-2050 innovation beyond today's technology options may occur in some areas). We have termed the scenario "MFR Midcentury" to better match the reviewer's description, and added the text below. We have also specified these example scenarios in Table 3 as well as in the corresponding Figures.

*L263-267: Our approach helps assess the potential of future innovation over the next eight decades beyond today's best available technologies, in the case of MFR. We also incorporate the possibility of no such innovation, showing an "MFR Midcentury" scenario that limits pollution control to the 2050 levels in GAINS. (Other studies could explore other scenarios based on the research question; we describe examples in the discussion and Table 3).*

Specific comments

L30: I would suggest adding some evidence from any epidemiological study, such as the latest GBD study (Murray et al 2020).

We have added the reference at the end of that sentence.

L35: Beyond its damages to crops, O3 has also a significant impacts on human health (e.g., Turner et al, 2016), particularly relevant for future scenario simulation (due to the large uncertainty on its precursors). It could be mentioned there.

We now include citations of $O_3$ health impacts as well as crop damage:

*L36-37: ...while ground-level ozone can increase mortality risk, exacerbate crop loss, and worsen socioeconomic disparities (Saari et al., 2017; Turner et al., 2016; Sampedro et al., 2020a).*

L85: Apart from full CTMs, the outcomes from TAPS could also be combined with air quality emulators to explore the concentration levels and health inputs of alternative scenarios, as it has been done in some of the studies mentioned in the introduction (e.g., Markandya et al, 2018; Vandyck et al, 2020; Reis et al 2022).

We agree, and have made a combined edit to address a similar note from the other reviewer.

L87-90: *In addition, its emissions outputs can provide flexibility for different air quality and health analyses – whether using emulators for rapid scenario study, or driving global atmospheric chemical transport models (CTMs) that avoid emulators' precalculated emissions-to-impact relationships.*

L130: Apart from the study, citing the CEDS release may be helpful to identify the version: (perhaps  https://doi.org/10.5281/zenodo.3865670)

We have added this link after confirming that it matches the dataset used.

L198: It would be useful to clarify why it is assumed that the MFR EFs tend to zero in those sectors where no activity (only emissions) is represented (e.g., waste).

MFR EFs trend to zero in the waste sector to maintain consistency with a recent GAINS paper (Gomez Sanabria et al., 2021), not because no activity (only emissions) is represented. The paper describes an MFR scenario to eliminate waste emissions of the studied air pollutant precursors by using circular waste strategies to avoid open burning. We have edited the text to clarify this:

L212-215: *For the GAINS waste sectors – where only emissions (not activities) were given – we assume constant emissions intensities for CLE, and follow a recent GAINS paper on MFR's elimination of open burning (Gomez Sanabria et al., 2021) to apply region-specific trends to zero by 2050 for MFR (based on MFR/CLE emissions ratios).*

L275: In Figure 2, adding scenario-specific lines within ranges (at least for TAPS) would be interesting to see the differences across the air pollution control strategies.

We have now added scenario-specific lines for TAPS in Figure 2 for clarity.

L319: It is strange that figures are shown up to 2050, but the results on this section 3.2. are discussed for the whole century. This should be consistent, either by adding 2100 figures (in addition of the results for 2050) or by discussing the effects until mid-century.

We have added new figures for 2100 (Figure 4 for sectors; Figure 6 for regions).

L320: Why is coal not phased-out with the implementation of the NDCs? This may be a result nfrom the EPPA model but could be additionally elaborated because it seems to be counterintuitive (also in L327).

As of EPPA's scenario definition in March 2021, most NDCs fail to phase out (or even phase down) coal. Table 3 of Paltsev et al. (2021) provides more detail between each region's trajectory and its countries' NDCs to 2030 – noting little phase-out in parts of Asia, Africa, Eastern Europe, and Latin America. We have aimed to clarify this point in the text below:

*L356-357: These regions also generally have NDCs (as of March 2021) that fail to phase down coal, according to Table 3 of Paltsev et al. (2021).*

L389: I do not think that a "constant EFs" scenario would be useful, considering that there is "current legislation" (CLE).

We agree and have removed that suggested scenario from the table.

L426: Beyond health impacts, the outcomes from TAPS combined with other tools (as for health) could be applied to explore other air-pollution-related damages on crops, biodiversity, forestry, or labor productivity.

We have included those ideas in the concluding text below.

*L474-478: Finally, additional research with air quality and impact models can assess the health, economic, and ecological effects of TAPS emissions scenarios as well as their implications for decision-making. Quantified impacts could include a range of mortality and morbidity endpoints to reflect recent epidemiological research (Danesh Yazdi et al., 2019), as well as other vulnerabilities (such as crops, biodiversity, and forestry) or analyses of equity, uncertainty, and sensitivity for key parameters (Hess et al., 2020).*

Anonymous Referee #2

**Review comments**

In this manuscript, the authors build a framework to study the long-term emissions of GHGs and major air pollutants by coupling with other socioeconomical models. This tool provides flexible to study the wide range of emission changes under various climate scenarios. The methodology is reasonable, and potential be useful. However, I have several concerns on how the authors solve their discrepancies between their framework with SSPs, and how the other researchers could be beneficial from this tool.

We appreciate the detailed review and questions. Throughout the paper, we have sought to clarify our intent and relationship with the SSPs. As Referee #1's comments have helped us to further clarify, our goal was not just to build more scenarios of long-term air pollutant emissions, but to enable the flexible study of how diverse climate and air quality actions could simultaneously affect such emissions. This goal is thus inherently different from that of the SSPs, which assess specific socioeconomic narratives and a specific air pollution reduction ambition within each. Given these important questions, we have added further detail in the text on the

tool's differences from the SSPs and the additional benefits of our approach in combination with existing tools, as noted under the second methods comment and elsewhere.

**Methods:**

To make this tool usable or accessible for other users, I want the authors to add their suggestions on how to include the fire & aviation emissions which are currently not included in this framework (line 142-144).

We appreciate the desire to make this tool more accessible and complete. We have included some suggestions in the text, including GFED for fire emissions and AEIC for aviation.

*L154-160: Other fire emissions could be added from GFED or similar inventories after deciding on their future trajectories (which we leave to later work, given large uncertainties). In addition, we do not currently include aviation emissions, given their exclusion from both $CEDS_{GBD-MAPS}$ and GAINS. Air pollution from aviation has been linked to 16,000 annual deaths (Eastham and Barrett, 2016), or less than 1% of pollution's estimated global mortalities (Murray et al., 2020). However, future efforts could consider sources such as the 2019 version of the Aviation Emissions Inventory Code (Simone et al., 2013), as used in GEOS-Chem (GEOS-Chem, 2021).*

As acknowledged in their discussions, the authors also discussed the discrepancies between TAPS and SSPs. To make community adopt their framework for further climate and air quality related research, I wonder how the authors can convince the users that these discrepancies between TAPS and SSPs are acceptable.

As mentioned above, differences between future scenarios such as the SSPs and our approach are expected given the different research purposes of TAPS and the SSPs. Our comparison to the SSPs should not be interpreted as direct model evaluation, but rather as an effort to provide information to readers who would like to understand differences between our model and the commonly-used SSP scenarios. We realize on review that this had not been clearly communicated, and have made efforts to improve this in the revised manuscript.

*L291-293: We also compare this implementation to corresponding SSP IAM and CMIP6 scenarios (summarized in Table 4), which serve a different research purpose than TAPS (and thus are not expected to match) but can act as a useful reference point.*

*L343-347: In the TAPS example policy scenarios, emissions often trend much higher if activity and intensity reductions are limited to current legislation. This result differs from the SSPs, which include actions beyond current legislation to answer different research questions (Rao et al., 2017). While recent studies support cases of increased emissions under current legislation (Rafaj et al., 2021), they focus on trends to mid-century.*

*L403-411: Several factors can help explain the different projection scenarios of TAPS and the SSPs. Most importantly, the two scenario sets serve different research goals – as the SSPs specify future pollution controls in line with each socioeconomic pathway (versus our broader range of outcomes). In practice, this leads to SSP emission factors that may trend much lower*

*than CLE, according to Table 1-2 and Fig. 1-1 of the Supporting Information in Rao et al. (2017). The resulting scenarios often have lower emissions than our "CLE Continued Trend", as well as other studies of GAINS CLE with the CMIP6 IAMs (Rafaj et al., 2021). The relevance of each scenario set will depend on the question at hand – and whether the underlying research question users seek to address can be answered by applying a specific socioeconomic pathway, or would benefit from a framework such as TAPS in which assumptions about air pollution are decoupled.*

**Minor comments:**

Line 32: delete "anthropogenic".

Done.

Line 40-41: I suggest the authors to expand their references on co-benefits analysis. These two included in the paper are not very representative.

We agree that more context is needed and now include several additional references, focusing on studies of air quality co-benefits from GHG emission reductions.

*L43-44: Gallagher and Holloway, 2020; Karlsson et al., 2020; Nemet et al., 2010; Rao et al., 2016; Sampedro et al., 2020b).*

Line 83-85: "dependence on previous CTM runs and base years"

We have made a combined edit to address a similar note from the other reviewer.

*L87-90: In addition, its emissions outputs can provide flexibility for different air quality and health analyses – whether using emulators for rapid scenario study, or driving global atmospheric chemical transport models (CTMs) that avoid emulators' precalculated emissions-to-impact relationships.*

Line 90: the authors discussed that the usages of EPPA are quite "limited" in previous studies. I wonder then why the authors choose the EPPA not other IAMs? Any potential limitations for potential users to adopt this product?

This was a miscommunication on our part. It is not EPPA in general, but EPPA's internal air pollutant emissions estimates that have "limited use". We have revised the text to be more specific. EPPA's economic and energy outputs have been widely used in climate-relevant studies, as mentioned in L93-94 and reviewed in Faehn et al. (2020). We follow those studies by using EPPA's available and relevant scenario data, though future studies could readily apply the TAPS framework to other IAMs after integrating the proper data (L127-130). We describe some limitations in L426-432.

*L96-100: While prior efforts have sought to endogenize EPPA's air pollutant emissions trends based on the cost of pollution control options (Sarofim, 2007; Valpergue De Masin, 2003;*

*Waugh, 2012), these internal estimates have been limited to select studies (Nam et al., 2013). In contrast, the TAPS framework combines EPPA's energy and land use outputs with other data to produce its own pollutant emissions scenarios, allowing it to be exercised autonomously for flexible scenario development (Fig. 1).*

Line 94: describe "EPPA7"

We have added description of EPPA7 (including a new reference to a recent model update) around the previous description of EPPA.

*L91-95: We demonstrate the tool with illustrative scenarios after coupling with the Economic Projection and Policy Analysis model version 7 (EPPA7). EPPA is a global multi-region multi-sector recursive–dynamic computable global equilibrium (CGE) model that has been used to study a variety of climate and economic policy impacts (Chen et al., 2015, 2017; Paltsev et al., 2005). EPPA7 is a recent version that includes updated economic data as well as new representations of advanced energy technologies (Chen et al., 2022).*

Line 125: there is no link or doi for such reference "GEOS-Chem, 2021".

We have added a link to that reference, as well as two others (from IIASA) that had previously been included without links.

Line 142: I am very curious to see the authors' check on the consistency between EPPA7 and GFED.

We have edited the text to specify that the consistency check was between different years of the GFED inventory, not between GFED and EPPA7. EPPA7 was only used to decide which base year to use (2014), and then to scale the GFED and CEDS inventories by future activities from agricultural waste burning and other emissions sources. GFED's 2014 agricultural waste burning emissions are generally consistent with the surrounding decadal average, as can be found at the SSP database (https://tntcat.iiasa.ac.at/SspDb/) or the current GFED portal (https://www.globalfiredata.org/).

*L150-152: We use 2014 values to match the base year of EPPA7; 2014 GFED emissions are generally consistent with emissions quantities from neighboring years.*